# Assessing knowledge about hypertension and identifying predictors of inadequate knowledge in Saudi Arabia: A cross-sectional study

Ajiad Alhazmi[1]*, Hassan N. Moafa[1,2], Mohammed Kotb[1¤a], Louay Sayegh[1],
Hassan Baydhi[1], Abdullaziz Hazzazi[1], Hassan Moafa[3¤b], Abdulelah Hakami[4]

1 Department of Epidemiology, College of Public Health and Tropical Medicine, Jazan University, Jazan, Saudi Arabia, 2 Department of Tropical Medicine, College of Public Health and Tropical Medicine, Jazan University, Jazan, Saudi Arabia, 3 Department of Health Education and Promotion, College of Public Health and Tropical Medicine, Jazan University, Jazan, Saudi Arabia, 4 Alhoma Primary Health Care, Western Sector, Saudi Arabia Ministry of Health, Jazan, Saudi Arabia

☯ These authors contributed equally to this work.
¤a Current Address: Department of Community Medicine, College of Medicine, Ain Shams University, Cairo, Egypt
¤b Current Address: Almaarefa University, Riyadh, Saudi Arabia
* ajiad.haz@gmail.com

**Data Availability Statement:** All relevant data are within the manuscript and its Supporting information files.

## Abstract

### Background

Globally, hypertension is among the leading causes of premature mortality. It is a noncommunicable disease characterized by a persistent state of raised blood pressure that increases the risk of cardiovascular diseases and medical conditions affecting the brain and kidneys. There is a paucity of thorough hypertension knowledge assessment among hypertensive patients in the Jazan region of Saudi Arabia. Thus, this study aimed to assess overall and specific knowledge about hypertension and to identify predictors of inadequate knowledge.

### Methods

A cross-sectional study was conducted in the Jazan region of Saudi Arabia between February and April 2023. Data were collected using an online, self-administered questionnaire divided into two sections. In the first section, the characteristics of the participants were collected. In the second section, the Hypertension Knowledge-Level Scale was used to measure overall and specific knowledge areas (subdimensions). The overall and subdimensional means were tested using Mann–Whitney U and Kruskal–Wallis H tests. Furthermore, the binary logistic regression was conducted to determine inadequate knowledge predictors.

### Results

In all 253 hypertensive patients were eligible for participation; almost 70% of whom were male. The mean age of the participants was 45 years (±14.7), and their mean overall

**Funding:** The authors received no specific funding for this work.

**Competing interests:** The authors have declared that no competing interests exist.

knowledge score was 17.60 (±5.09), which was equivalent to 67.7% of the maximum score. In addition, 40.7% of participants had an adequate level of hypertension knowledge. The complications subdimension level of knowledge was borderline optimal. At the same time, an adequate knowledge level was detected only in the lifestyle subdimension.

## Conclusion

Most patients showed inadequate levels of knowledge related to hypertension management. Diet, medical treatment, disease definition, drug compliance, and complications were subsequently the least knowledgeable subdimensions among the study population. Therefore, these subdimensions should be prioritized when planning hypertension educational interventions and during follow-up sessions, especially for patients of younger age groups and those with lower educational levels.

## Introduction

One of the major causes of global premature mortality is hypertension (high blood pressure). It is defined either through specific measurements of blood pressure or through the reported use of antihypertensive medications. It is a noncommunicable disease characterized by persistently elevated blood pressure. High blood pressure is a serious medical condition that increases the risk of cardiovascular diseases and affects the brain and kidneys [1].

In a 2019 study, the global, age-standardized prevalence of hypertension was found to be 34% for men and 32% for women aged 30–78 years. These findings were similar to the levels found in 1990, when the prevalence was 32% for both men and women. While developed countries have experienced a decline in the prevalence rate of hypertension, it has increased in developing countries, where nearly two-thirds of hypertensive patients live [2]. Given that hypertension typically presents with no symptoms, an estimated 46% of patients are unaware of their condition. However, symptoms such as headache, blurred vision, and chest pain can appear in instances of extremely high blood pressure levels [1]. According to a study regarding the burden of cardiovascular disease risk factors, the prevalence rate of hypertension in Saudi Arabia in 2017 for people above 18 years was 31.4% for Saudis and 47.5% for expatriates who reside in Saudi Arabia [3]. During the same year, the Saudi Ministry of Health documented that the prevalence rate of hypertension reached 3.2% among people aged 15 to 24 years, 51.2% among those aged 55 to 64 years, and up to 70% among people aged 65 years or above. Moreover, the overall prehypertensive prevalence rate was 40.4% [4]. In particular, the prevalence rate of hypertension in the Jazan region reached 8.9% [5]. More importantly, the prevalence rate of hypertension among Saudis is predicted to increase along with the global trend [6]. As a result, the incidence of complications resulting from uncontrolled hypertension will subsequently increase.

Many risk factors can increase the risk of developing hypertension. All these factors can be attributed to two types. The first type is modifiable factors, including an unhealthy diet, physical inactivity, smoking, alcoholism, obesity, and comorbidities such as diabetes mellitus and kidney disease. The second type is uncontrollable nonmodifiable risk factors, such as family history and age [1]. Adequate knowledge of these factors and nonpharmacological approaches can positively influence hypertension management, as numerous studies have shown a positive relationship between knowledge and management of the disease [7–12]. Given that knowledge

of hypertension management plays a significant role in prevention and control, research investigating this knowledge in hypertensive patients is vital for public health.

Several studies have assessed hypertension knowledge among hypertensive patients using the Hypertension Knowledge Level Scale (HK-LS). For example, a Palestinian study revealed that the overall mean knowledge level was equivalent to 82.8% of the maximum score [13]. A Saudi study found that the overall knowledge level was equivalent to 70.2% of the maximum scores, and the mean knowledge level differed significantly according to sex, age, educational level, and employment status [14]. A recent study conducted in various regions of Saudi Arabia showed that the overall mean knowledge level was equivalent to 68.5% of the maximum score, and the least knowledgeable subdimensions were related to drug compliance followed by diet. In contrast, the complications and lifestyle subdimensions were optimal [15]. These previous studies investigating hypertensive patients' knowledge levels in other Saudi regions cannot be generalized to other Saudi regions, such as the Jazan region, where a higher proportion of people suffer from hypertension and its complications and where the health system is overwhelmed by emergencies and cases of communicable and genetic diseases. In addition, this region has a fairly different health system compared with other Saudi regions, particularly in terms of structural indicators [16–18].

As a result of the limited data, there is a need to understand the knowledge in general and specific areas concerning hypertension in this particular region of Saudi Arabia. Therefore, this study aimed to assess the levels of overall and specific knowledge of hypertension and to identify the predictors of inadequate levels of knowledge in hypertensive patients in the Jazan region of Saudi Arabia. The results of this study could be used as baseline data for prospective studies and actions to improve hypertension management in the region.

## Materials and methods

### Study design and settings

A cross-sectional study was conducted using an online, self-administered questionnaire disseminated to hypertensive patients in the Jazan region from 5 February to 20 April 2023. The region is located in southwestern Saudi Arabia and comprises 13 provinces distributed in a geographical area of 13,400 km$^2$, the region's total area. The regional population is 1.6 million, and most live in rural areas [19]. Areas of less than 5,000 inhabitants can be defined as rural areas, while geographical areas with more than 5,000 inhabitants can be defined as urban [20, 21]. In Saudi Arabia, health care services are free of charge for all Saudi citizens, while for non-Saudis, the services are provided via health insurance companies. Health education services are available at all primary health care centers and are provided only by the family physicians working there. Most of the health care services in the Jazan region are operated mainly by the Ministry of Health, in addition to a limited number of private hospitals and medical complexes. The Jazan region has 177 primary health care centers and 284 family physicians [16, 22]. There was a ratio of 1.8 family physicians per 10,000 population in 2021 in the Jazan region. Patients with essential hypertension are screened and treated at primary health care centers, while those with secondary hypertension due to other medical conditions or who have hypertension complications are referred to secondary health care services represented by public hospitals.

### Study participants and eligibility criteria

The sample size was determined using a previously suggested formula (n = z$^2$pq/d$^2$) [23]. It was calculated with a (95%) confidence interval and a (6%) margin of error. Due to a lack of previous studies in the Jazan region, we used the proportion of those with inadequate

hypertension knowledge based on the same criteria of knowledge level classification from other regions of Saudi Arabia, which was 61.3% [24–27]. A total of 253 subjects were subsequently needed.

The eligibility criteria were clarified on the first page of the online version of the survey along with the study objectives, estimated time for completing the questionnaire and notification of voluntary participation in addition to the right to withdraw from the study at any time. The criteria for participation included the following: Saudi citizenship; residence in the Jazan region; use of antihypertensive medications within the last three months; at least 18 years of age; and willingness to participate in the study, indicated by the provision of informed consent. Individuals who did not meet at least one criterion were automatically excluded. The online version of the survey was disseminated through WhatsApp groups, a popular social media platform in Saudi Arabia, with a request to forward the survey to other groups. Data were collected anonymously. This study complies with Strengthening the Reporting of Observational Studies in Epidemiology guidelines, as shown in S1 Table [28].

### Data collection instrument

Study data were collected using an online, self-administered, closed-ended questionnaire developed via Google Forms. The questionnaire was divided into two sections. In the first section, data related to the characteristics of the participants, including sex, age, educational level, residency area, history of hypertension complications and time since diagnosis, were collected. Educational level was categorized according to the Saudi educational system, comprising essential and higher education. Essential education includes primary, intermediate and secondary schools. Higher education includes graduate and postgraduate studies. Participants without education at any of these levels but who could read and write were referred to as having no formal education. Participants were requested to report whether they had a history of hypertension complications and were then categorized as having either a negative or positive history. In addition, those who reported any form of complication were referred to as having a positive history of complications and were further divided according to the type of complication. Those with two or more complications were recorded as having multiple complications. The second section used the 26 items of the HK-LS instrument, which was developed by Erkoc et al. in English to assess Turkish adults' knowledge about hypertension and its management using 22 items that are distributed across six specific areas called subdimensions, namely: disease definition, medical treatment, drug compliance, lifestyle, diet, and complications [29]. Due to the validity and reliability of the instrument in Saudi Arabia, [15] the items were translated into Arabic. However, certain modifications were applied to the scale after consultation with experts, including an addition of one item to the disease definition subdimension, and it was added to assess participants' knowledge regarding the increased isolated systolic pressure as a form of hypertension. Also, three items were added to the lifestyle subdimension to evaluate patients' understanding of the role of salt intake, obesity/overweight, and physical activity as essential factors for managing hypertension. The 26 items of the scale were distributed over the six subdimensions as the following: disease definition (three items); medical treatment (four items); drug compliance (three items); lifestyle (eight items); diet (two items); and complications (five items). Each item involved a complete sentence, and the participant was requested to specify whether the given statement was 'true', 'false' or 'do not know'. Each correct answer was worth one point. Incorrect or 'do not know' answers received zero points. Study participants were classified as having an adequate or inadequate level of knowledge based on the number of correct responses. An adequate level of knowledge required at least 75% of the responses to be correct (20 or more out of 26 questions). If the number of correct

responses was less than 75% (fewer than 20 out of 26), the participant was classified as having inadequate knowledge.

## Pilot study

A pilot study was performed to assess the modified subdimensions. This phase of the study consisted of 13 participants who did not participate later in the study. The modified subdimensions were the disease definition and lifestyle subdimensions. The internal consistency of these modified subdimensions was assessed using Cronbach's alpha, and the results indicated the reliability of these subdimensions S2 Table.

## Ethical considerations

The Committee of Scientific Research at Jazan University issued ethical approval for this study on February 5, 2023 (reference number: REC-44/07/548). Written informed consent was provided by all subjects. The study was carried out in accordance with the Declaration of Helsinki, 2013. Once data were collected, they were kept secured, and only the investigators of the study had the right to access them.

## Statistical analysis

Frequency and percentage distributions were calculated for categorical variables. Means and standard deviations were measured for the overall and subdimensional results of the HK-LS. Data were not normally distributed S3 Table. Therefore, nonparametric tests such as the Mann–Whitney U test and Kruskal–Wallis H test were applied to explore significant knowledge differences according to the characteristics of the participants at the overall and subdimensional levels of the HK-LS. The mean ranks are presented in S4 Table. The predictors of inadequate knowledge were identified through binary logistic regression. Cohorts with the highest means at the overall level were chosen as referent categories. For all tests, $P < 0.05$ was considered significant. Statistical analysis was performed using SPSS version 23 (IBM Corp., Armonk, NY, USA).

# Results

## Characteristics of hypertensive participants

Among a total of 634 respondents, only 253 were eligible. The rest were ineligible, and the response rate was 39.9%. Table 1 shows the characteristics of eligible participants. The total number of participants was 253; most were male (60.9%). The mean age was 46.7 (±14.7) years and ranged from 22 to 90 years. Of the total, 52.6% were from rural areas, and 41.5% had achieved higher education. We found that 32.0% of participants had hypertension for less than two years. In addition, the predominant two types of complications were mainly related to the eyes (21.7%), followed by heart-related complications (12.3%). Only 40.7% of participants were found to have an adequate level of knowledge about hypertension.

## Overall HK-LS scores

Table 2 shows the overall mean HK-LS scores according to the characteristics of hypertensive participants. The overall mean HK-LS score was 17.60, which was equivalent to 67.7% of the maximum score. We found that the mean HK-LS score significantly increased as age and educational level increased ($P = 0.022$ and $P < 0.001$, respectively). We found a significant difference in the mean HK-LS score between participants with a positive and negative history of complications, with a mean score of 18.34 and 16.81 in those who did and did not develop hypertension

**Table 1. Characteristics of the studied hypertensive participants.**

| Variables | n/mean | %/SD |
|---|---|---|
| **Sex** | | |
| Male | 154 | 60.9% |
| Female | 99 | 39.1% |
| **Age** | | |
| Mean age (±SD) in years | 45 | (±5.1) |
| **Age group** | | |
| <35 years | 57 | 22.5% |
| 35–44 years | 72 | 28.5% |
| 45–54 years | 42 | 16.6% |
| 55–64 years | 51 | 20.1% |
| ≥65 years | 31 | 12.3% |
| **Educational level** | | |
| No formal education | 50 | 19.7% |
| Primary | 17 | 6.7% |
| Intermediate | 46 | 18.1% |
| Secondary | 32 | 12.6% |
| Higher education | 108 | 42.5% |
| **Residency area** | | |
| Urban | 120 | 47.4% |
| Rural | 133 | 52.6% |
| **History of complications** | | |
| Negative | 122 | 48.2% |
| Heart complications | 31 | 12.3% |
| Eye complications | 55 | 21.7% |
| Kidney complications | 17 | 6.7% |
| Multiple complications | 28 | 11.1% |
| **Duration since diagnosis** | | |
| Since less than 2 years | 81 | 32.0% |
| Since 2–5 years | 63 | 24.9% |
| Since 6–10 years | 64 | 25.3% |
| Since more than 10 years | 45 | 17.8% |
| **Level of knowledge** | | |
| Adequate | 103 | 40.7% |
| Inadequate | 150 | 59.3% |

SD: Standard deviation.

complications, respectively ($P = 0.017$). We found a nonsignificant difference in the mean HK-LS score between female participants (18.26) and male participants (17.18) ($P = 0.100$).

## HK-LS subdimension scores

Table 3 shows the mean scores for all HK-LS subdimensions according to the characteristics of the participants. The overall mean scores across the different subdimensions were 1.74 for disease definition, 2.23 for medical treatment, 2.53 for drug compliance, 6.45 for lifestyle, 1.09 for diet, and 3.57 for complications. The lifestyle subdimension had the highest mean score, while the diet subdimension had the lowest. Notably, the means for certain HK-LS subdimensions

**Table 2. Overall mean HK-LS scores according to characteristics of hypertensive participants.**

| Variables | Overall HK-LS score (Mean±SD) | *P* value |
|---|---|---|
| | Min–Max 0–26 | |
| **All participants (N = 253)** | 17.60±5.09 | - |
| **Sex** | | |
| Male | 17.18±5.53 | 0.234[a] |
| Female | 18.26±4.29 | |
| **Age group** | | |
| <35 years | 16.05±5.19 | 0.022[b] |
| 35–44 years | 17.40±4.71 | |
| 45–54 years | 17.95±5.36 | |
| 55–64 years | 18.35±5.47 | |
| ≥65 years | 19.23±3.71 | |
| **Educational level** | | |
| No formal education | 14.52±6.19 | <0.001[b] |
| Primary | 16.47±5.29 | |
| Intermediate | 17.52±4.45 | |
| Secondary | 17.75±5.52 | |
| Higher education | 19.20±4.24 | |
| **Residency area** | | |
| Urban | 17.46±5.25 | 0.705[a] |
| Rural | 17.74±4.97 | |
| **History of complications** | | |
| Negative | 16.81±5.54 | 0.029[a] |
| Positive | 18.34±4.55 | |
| **Duration since diagnosis** | | |
| Since less than 2 years | 17.09±5.03 | 0.535[b] |
| Since 2–5 years | 17.59±5.48 | |
| Since 6–10 years | 17.78±5.26 | |
| Since more than 10 years | 18.31±4.46 | |

SD: Standard deviation, Min: Minimum score, Max: Maximum score.

[a]Mann–Whitney U test,

[b]Kruskal–Wallis H test.

varied according to different participant variables. However, neither lifestyle nor diet subdimensions revealed a significant relationship with participant variables.

## Predictors associated with inadequate knowledge

The predictors of inadequate knowledge are listed in Table 4. The significant predictors of inadequate knowledge were younger age groups and lower educational levels in the crude and adjusted models. In addition, those who had been diagnosed with hypertension for less than two years were more likely to have an inadequate level of knowledge (COR: 2.418; 95% CI: 1.143–5.111; ($P$ = 0.021)).

## Discussion

Our study showed remarkable differences in the HK-LS score in a special Saudi region that has the highest population density in comparison with the rest of the regions in Saudi Arabia.

**Table 3. Subdimensions of the HK-LS according to the characteristics of hypertensive participants.**

| Variables | HK-LS Subdimensions (Mean±SD) | | | | | |
|---|---|---|---|---|---|---|
| | **Disease Definition** | **Medical Treatment** | **Drug Compliance** | **Lifestyle** | **Diet** | **Complications** |
| | **Min–Max** | **Min–Max** | **Min–Max** | **Min–Max** | **Min–Max** | **Min–Max** |
| | **0–3** | **0–4** | **0–4** | **0–8** | **0–2** | **0–5** |
| **Sex** | | | | | | |
| Male | 1.71±1.31 | 2.05±1.23 | 2.55±1.34 | 6.27±2.00 | 1.11±0.81 | 3.49±1.74 |
| Female | 1.77±1.22 | 2.51±1.04 | 2.52±1.16 | 6.73±1.55 | 1.05±0.80 | 3.70±1.58 |
| P value[a] | 0.778 | 0.004 | 0.435 | 0.075 | 0.556 | 0.403 |
| **Age group** | | | | | | |
| <35 years | 1.60±1.29 | 1.89±1.22 | 2.18±1.42 | 6.00±2.00 | 0.89±0.72 | 3.49±1.68 |
| 35–44 years | 1.75±1.24 | 2.13±1.14 | 2.44±1.31 | 6.44±1.78 | 1.06±0.79 | 3.58±1.77 |
| 45–54 years | 1.57±1.35 | 2.31±1.30 | 2.62±1.19 | 6.52±1.92 | 1.29±0.81 | 3.64±1.65 |
| 55–64 years | 1.98±1.23 | 2.37±1.15 | 2.88±1.11 | 6.61±1.94 | 1.10±0.86 | 3.41±1.82 |
| ≥65 years | 1.77±1.33 | 2.74±0.93 | 2.71±1.16 | 6.90±1.22 | 1.23±0.85 | 3.87±1.23 |
| P value[b] | 0.550 | 0.012 | 0.084 | 0.185 | 0.119 | 0.954 |
| **Educational level** | | | | | | |
| No formal education | 1.30±1.33 | 1.78±1.15 | 1.92±1.32 | 5.72±2.56 | 0.86±0.78 | 2.94±1.91 |
| Primary | 1.18±1.19 | 1.65±1.27 | 2.47±1.18 | 6.12±2.12 | 1.24±0.83 | 3.82±1.38 |
| Intermediate | 1.59±1.31 | 2.39±0.95 | 2.43±1.41 | 6.50±1.55 | 1.09±0.76 | 3.52±1.66 |
| Secondary | 1.69±1.26 | 2.34±1.26 | 2.59±1.13 | 6.50±1.74 | 0.97±0.82 | 3.66±1.72 |
| Higher education | 2.10±1.17 | 2.43±1.19 | 2.85±1.15 | 6.80±1.43 | 1.20±0.81 | 3.82±1.55 |
| P value[b] | <0.001 | 0.005 | <0.001 | 0.158 | 0.103 | 0.070 |
| **Residency area** | | | | | | |
| Urban | 1.69±1.31 | 2.13±1.29 | 2.42±1.36 | 6.47±1.81 | 1.07±0.80 | 3.68±1.65 |
| Rural | 1.77±1.25 | 2.32±1.08 | 2.64±1.19 | 6.43±1.88 | 1.11±0.81 | 3.47±1.70 |
| P value[a] | 0.609 | 0.304 | 0.326 | 0.955 | 0.690 | 0.288 |
| **History of complications** | | | | | | |
| Negative | 1.66±1.32 | 2.08±1.16 | 2.48±1.26 | 6.30±1.90 | 1.07±0.85 | 3.23±1.86 |
| Positive | 1.81±1.24 | 2.37±1.19 | 2.59±1.29 | 6.59±1.78 | 1.10±0.76 | 3.89±1.43 |
| P value[a] | 0.359 | 0.045 | 0.316 | 0.117 | 0.864 | 0.005 |
| **Duration since diagnosis** | | | | | | |
| Since less than 2 years | 1.64±1.26 | 2.06±1.18 | 2.31±1.38 | 6.27±1.72 | 1.10±0.75 | 3.70±1.68 |
| Since 2–5 years | 1.89±1.27 | 2.10±1.15 | 2.71±1.14 | 6.48±1.92 | 1.05±0.85 | 3.37±1.81 |
| Since 6–10 years | 1.75±1.25 | 2.36±1.24 | 2.64±1.21 | 6.61±1.89 | 1.13±0.81 | 3.30±1.69 |
| Since more than 10 years | 1.67±1.38 | 2.53±1.12 | 2.53±1.31 | 6.49±1.90 | 1.07±0.84 | 4.02±1.37 |
| P value[b] | 0.710 | 0.062 | 0.359 | 0.226 | 0.961 | 0.074 |
| **Total** | 1.74±1.27 | 2.23±1.18 | 2.53±1.27 | 6.45±1.84 | 1.09±0.80 | 3.57±1.67 |

SD: Standard deviation, Min: Minimum score, Max: Maximum score.

[a]Mann–Whitney U test,

[b]Kruskal–Wallis H test.

Interestingly, to the best of our knowledge, our study is the only Saudi study to assess the overall and subdimensional knowledge of hypertension and identify predictors of inadequate knowledge in the Jazan region.

We found that the overall mean score of the knowledge levels of the participants was 17.60 (±5.09), which represents 67.7% of the maximum score. The majority of participants (59.3%); (95% CI: 53.0%-66.0%) were classified as having inadequate knowledge about the disease and

**Table 4. Independent predictors of inadequate knowledge.**

| Predictors | Crude results | | | | Adjusted results | | | |
|---|---|---|---|---|---|---|---|---|
| | **B** | **OR** | **OR (95% CI)** | ***P* value** | **B** | **OR** | **OR (95% CI)** | ***P* value** |
| **Sex** | | | | | | | | |
| Male | 0.116 | 1.123 | 0.673–1.876 | 0.657 | 0.226 | 1.253 | 0.682–2.301 | 0.467 |
| Female | | | Referent | | | | Referent | |
| **Age group** | | | | | | | | |
| <35 years | 1.266 | 3.548 | 1.417–8.885 | 0.007 | 2.505 | 12.246 | 3.359–44.639 | <0.001 |
| 35–44 years | 1.082 | 2.950 | 1.238–7.031 | 0.015 | 2.497 | 12.141 | 3.579–41.178 | <0.001 |
| 45–54 years | 0.421 | 1.523 | 0.598–3.882 | 0.378 | 1.474 | 4.368 | 1.332–14.323 | 0.015 |
| 55–64 years | 0.286 | 1.331 | 0.541–3.275 | 0.533 | 0.811 | 2.251 | 0.814–6.226 | 0.118 |
| ≥65 years | | | Referent | | | | Referent | |
| **Educational level** | | | | | | | | |
| No formal education | 1.120 | 3.065 | 1.468–6.399 | 0.003 | 1.347 | 3.847 | 1.723–8.589 | 0.001 |
| Primary | 1.253 | 3.500 | 1.073–11.419 | 0.038 | 2.066 | 7.893 | 2.078–29.984 | 0.002 |
| Intermediate | 0.608 | 1.837 | 0.905–3.728 | 0.092 | 2.191 | 8.942 | 2.996–26.691 | <0.001 |
| Secondary | 0.454 | 1.574 | 0.707–3.503 | 0.266 | 1.674 | 5.332 | 1.896–14.995 | 0.002 |
| Higher education | | | Referent | | | | Referent | |
| **Residency area** | | | | | | | | |
| Urban | 0.056 | 1.058 | 0.640–1.748 | 0.827 | 0.052 | 1.053 | 0.593–1.868 | 0.860 |
| Rural | | | Referent | | | | Referent | |
| **History of complications** | | | | | | | | |
| Negative | 0.307 | 1.360 | 0.821–2.251 | 0.232 | 0.534 | 1.705 | 0.961–3.026 | 0.068 |
| Positive | | | Referent | | | | Referent | |
| **Duration since diagnosis** | | | | | | | | |
| Since less than 2 years | 0.883 | 2.418 | 1.143–5.111 | 0.021 | 0.229 | 1.257 | 0.494–3.197 | 0.631 |
| Since 2–5 years | 0.486 | 1.626 | 0.752–3.516 | 0.216 | 0.220 | 1.246 | 0.519–2.991 | 0.622 |
| Since 6–10 years | 0.449 | 1.566 | 0.727–3.374 | 0.252 | 0.376 | 1.457 | 0.619–3.430 | 0.389 |
| Since more than 10 years | | | Referent | | | | Referent | |

Regression coefficient, OR: Odds ratio, CI: Confidence interval.

its management. These findings at the overall level are relatively similar to those of a previous Saudi study conducted by Abualnaja OS et al. in Makkah and another recent study conducted by Alshammari SA et al. to assess the validity and reliability of the HK-LS, where they reported an overall knowledge score of 70.2% and 68.5%, respectively [14, 15]. On the other hand, many other studies conducted in the kingdom have utilized different instruments to assess knowledge. For instance, an overall mean knowledge score of 77.1% was found in the Riyadh region [30]. In the Makkah and Jeddah regions, the proportion of hypertensive patients with an adequate level of knowledge was 73.0% and 72.6%, respectively [24, 31]. In comparison, the lower mean knowledge score and number of people with an adequate level of knowledge observed in our study could be attributed to differences in the scoring systems used for assessing knowledge level, particularly the differences in assessment items.

At the subdimensional level, participants displayed adequate knowledge only in the lifestyle subdimension, as the mean was approximately 80.0% of the maximum score. The mean knowledge scores in other subdimensions were equivalent to 54.5% for diet, 58.0% for both disease definition and medical treatment, 63.3% for drug compliance, and 71.4% for complications. These results are still inadequate, especially considering that the subjects are

hypertensive patients. Compared with our study, the study by Abualnaja OS et al. documented similar findings for the disease definition and drug compliance subdimensions [14]. However, deviations were observed in the subdimensions of medical treatment, lifestyle, diet and complications. In comparing these four subdimensions, participants in our study were characterized by higher knowledge only in the lifestyle subdimension. These deviations might be attributed to the urbanity of the Makkah population, in contrast to the Jazan region, where a higher proportion of people live in rural areas [19]. In addition, an African systematic review revealed a rural setting is a factor that influences the knowledge of cardiovascular diseases [32]. Furthermore, other Saudi studies identified the lowest knowledge level in the diet section [24, 26, 33].

A Ghanian study identified educational level and marital status as significant predictors of knowledge; [34] however, in our study, the significant predictors were a younger age group and lower educational level. Differences in study inclusion criteria might be responsible for this divergence in results since the study sample only included people who live in rural communities. While a diagnosis of hypertension within the previous two years was found to be a significant predictor in the current study, this was only in the crude model, and when adjusted, it appeared insignificant.

Uncontrolled hypertension can lead to serious complications such as ischemic heart attack and cerebral stroke, which require facilities that are fully equipped with advanced tools in addition to the availability of highly skilled providers to intervene early and curb the possible sequelae that may arise. For coronary vascular disease requiring cardiac catheterization, there is only one health care facility, called Mohammad bin Nasser Hospital, which is overwhelmed by 9,168 cases already registered [17]. While stroke cases require advanced facilities to provide fibrinolytic medical intervention and rehabilitation facilities, the region lacks this type of facility, and therefore, those stroke patients also add an extra overwhelming burden to the health care system in terms of regional hospital beds and extra cost. Moreover, the region's health care system is burdened by cases of communicable and genetic diseases [16, 18]. All these factors might encourage policy-makers to draw attention to providing community-based health education that supports primary health care centers in educating patients. Knowledge about hypertension is a significant, independent contributor to successfully controlling the condition [7–12]. Unfortunately, social media platforms have been identified by many studies as being a significant source of misinformation that threatens public health by spreading incorrect health information and rumors about noncommunicable diseases, treatments, drugs, and lifestyles [35–38]. While there have been few studies on hypertension-related social media content, one study assessed hypertension content on YouTube and found that one-third contained misleading information [39]. Hence, to maximize community-level knowledge, health care providers should be patients' primary sources of knowledge.

Knowledge about hypertension and adherence to hypertension guidelines among physicians at primary health care centers in the Jazan region was recently investigated by Shnaimer JA et al. They assessed knowledge and adherence by adopting an assessment component from the Saudi Hypertension Management Guidelines. Strikingly, Shnaimer JA et al. revealed inadequate knowledge about hypertension among physicians, with an overall mean knowledge of 7.9 out of 16, equivalent to 49.4% of the maximum score. The mean score for adherence to hypertension guidelines was 7.11 out of 11, representing 64.6% of the maximum score. They also highlighted that the higher number of patients seen daily played a significant role in physicians' knowledge scores and guideline adherence [6]. The provision of health education exclusively through family physicians, the ratio of whom to the Jazan regional population was 1.8 per 10,000 population in 2021, creates an effort burden for those physicians in general practices to offer proper interventional health education and promotion. Increasing the number of

family physicians per 10,000 population could add extra costs, although it might improve patients' knowledge levels about different preventable diseases. Another solution is to involve health educators who require a shorter training period and lower costs than family physicians in educating patients about several preventable risk factors that affect the progress of hypertension and other diseases. Health education interventions have been shown to significantly affect hypertension control [40, 41] and could improve knowledge about hypertension in patients. This is especially needed in the Jazan region, where there is inadequate knowledge and adherence by primary health care physicians and a low family physician per population ratio.

This study has several limitations. First, it used a cross-sectional design, which provided insights regarding the community only at the time of the study. Second, the sampling technique and the online questionnaire preclude the presentation of the whole area and the generalizability of the results. Third, our data did not involve those patients suffering from severe complications, such as cerebral stoke, due to the difficulty of communication with them. Subsequently, future studies are required to determine their knowledge level. Nevertheless, our study used the HK-LS, and this work can be considered by other researchers as adding value to the evidence for understanding the status quo and conducting future research.

## Conclusion

Most patients showed inadequate levels of knowledge related to hypertension management. The inadequate level of knowledge was remarkable with young hypertensive patients and those with a lower level of education. In addition, the level of knowledge related to the subdimensions of diet, medical treatment, disease definition, drug compliance, and complications was inadequate. Therefore, to improve patients' knowledge and enhance public health practices in Jazan, involving public health educators in primary health care services could improve patients' knowledge about hypertension and its complications and reduce the burden on family physician burdens in the Jazan region, as indicated in the literature. Further studies are required to determine the knowledge level of hypertensive patients with severe complications such as cerebral stroke.

## Supporting information

**S1 File. Data.** Data of eligible participants.
(XLSX)

**S1 Table. STROBE checklist.** STROBE Statement—checklist of items that should be included in reports of observational studies.
(DOCX)

**S2 Table. Modification details.** Modifications applied to the HK-LS.
(DOCX)

**S3 Table. Tests of normality.** Tests of normality of dependent variables.
(DOCX)

**S4 Table. Mean ranks.** Mean ranks of nonparametric tests.
(DOCX)

## Author Contributions

**Conceptualization:** Ajiad Alhazmi, Hassan N. Moafa, Mohammed Kotb, Abdullaziz Hazzazi, Hassan Moafa.

**Data curation:** Ajiad Alhazmi, Hassan N. Moafa, Louay Sayegh, Hassan Baydhi, Hassan Moafa.

**Formal analysis:** Ajiad Alhazmi, Hassan N. Moafa, Louay Sayegh, Abdulelah Hakami.

**Investigation:** Ajiad Alhazmi, Hassan N. Moafa, Mohammed Kotb, Louay Sayegh, Hassan Baydhi, Abdullaziz Hazzazi, Hassan Moafa, Abdulelah Hakami.

**Methodology:** Ajiad Alhazmi, Hassan N. Moafa, Mohammed Kotb.

**Project administration:** Ajiad Alhazmi.

**Resources:** Ajiad Alhazmi, Hassan N. Moafa.

**Supervision:** Ajiad Alhazmi, Hassan N. Moafa, Mohammed Kotb.

**Validation:** Ajiad Alhazmi, Hassan N. Moafa, Mohammed Kotb.

**Visualization:** Ajiad Alhazmi, Hassan N. Moafa, Louay Sayegh, Abdulelah Hakami.

**Writing – original draft:** Ajiad Alhazmi, Hassan N. Moafa, Hassan Baydhi, Abdullaziz Hazzazi, Abdulelah Hakami.

**Writing – review & editing:** Ajiad Alhazmi, Hassan N. Moafa, Mohammed Kotb, Hassan Moafa.

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
