## [Decision Letter · Decision Letter 0]

12 Jan 2024

PONE-D-23-25700Assessing knowledge about hypertension and identifying predictors of inadequate knowledge in the Jazan regionPLOS ONE

Dear Dr. Alhazmi,

Thank you for submitting your manuscript to PLOS ONE. After careful consideration, we feel that it has merit but does not fully meet PLOS ONE’s publication criteria as it currently stands. Therefore, we invite you to submit a revised version of the manuscript that addresses the points raised during the review process.Required changes for acceptance. I have reviewed the reviewers' comments and outlined below those that need to be addressed before acceptance and those that are recommendations only. Title & Abstract Title: Indicate the type of study in the title.Background: state the gap(s) in the current understanding of hypertension in the Jazan region that informed the current studyMethods: Include the statistical analysis employedThe conclusion section of the abstract should be rephrased according to the main findings of the study (Reviewer 1)The sampling technique and the online questionnaire preclude the presentation of whole area and generalizability of the results. This must be added to the limitation section (Reviewer 1)IntroductionLine 84-92 – the author writes ‘These previous studies investigating hypertensive patients' knowledge levels in other Saudi regions cannot be generalized to other Saudi regions, such as the Jazan region, where a higher proportion of people suffer from hypertension and its complications and where the health system is overwhelmed by emergencies and cases of communicable and genetic diseases’ – can you conform whether study with numbered reference 15 (by  Alshammari et al) was conducted in other region of Saudi Arabia?  Or the sample was taken from the whole of Saudi Arabia? If the latter is true, then you will need to refine the statement of your evidence gap.Recommended changesNo need to mention the region of the study to enhance the global significant. You can add the Country name. All comments by reviewer 2 are recommendations only.

We look forward to receiving your revised manuscript.

Kind regards,

Isaac Amankwaa, Ph.D.

Guest Editor

PLOS ONE

Journal Requirements:

2. No need to ping with follow up.

3. We note that your Data Availability Statement is currently as follows: "All relevant data are within the manuscript and its Supporting Information files."

Reviewers' comments:

Reviewer's Responses to Questions

**Comments to the Author**

1. Is the manuscript technically sound, and do the data support the conclusions?

Reviewer #1: Yes

Reviewer #2: Yes

2. Has the statistical analysis been performed appropriately and rigorously? 

Reviewer #1: Yes

Reviewer #2: Yes

3. Have the authors made all data underlying the findings in their manuscript fully available?

Reviewer #1: Yes

Reviewer #2: Yes

4. Is the manuscript presented in an intelligible fashion and written in standard English?

Reviewer #1: Yes

Reviewer #2: Yes

5. Review Comments to the Author

Reviewer #1: The manuscript is technically sound, the data support the conclusions and presented in an intelligible fashion. However, minor revision is needed to complete the picture. In the title its very important to indicate the type of the study. No need to mention the region of the study to enhance the global significant. You can add the Country name. The conclusion section of the abstract should be rephrased according to the main findings of the study. The sampling technique and the online questionnaire preclude the presentation of whole area and generalizability of the results. This must be added to the limitation section.

Reviewer #2: It is well written and technically sound.No issues of dual publications and research ethics.Although it attempted to address this important topic there are some critical issues that warrant further attention. These would improve the research's quality and robustness, making it more academic journal-worthy. Points for improvement:

1. Title and Abstract: Text clarity, conciseness and structure can be improved.

2. Literature assessment: A more thorough assessment would strengthen the study's scientific foundation. This would help understand the present state of knowledge and the research gaps the project attempts to address.

3. Methods: More detailed information about the tool for data would be helpful.

4.Discussion: Attention and conciseness can be improved by emphasizing the most important facts and recommendations.

6. PLOS authors have the option to publish the peer review history of their article (what does this mean?). If published, this will include your full peer review and any attached files.

Reviewer #1: No

Reviewer #2: No

---

## [Author Response · Author response to Decision Letter 0]

5 Feb 2024

COVER LETTER

Manuscript reference number: PONE-D-23-25700

Title: Assessing knowledge about hypertension and identifying predictors of inadequate knowledge in the Jazan region

Dear Editor, 

We would like to thank the Editor for giving us the opportunity to revise the manuscript. The reviewers have some excellent remarks, which we have tried to incorporate in this revised version. Below, we describe point-by-point how we handled the comments from the reviewers on our previous submission. The changes are highlighted using track changes in the manuscript. Page numbers are referring to the document entitled as “Revised Manuscript with Track Changes”.

Editor comment:

Point (Pointed out within the Email that the corresponding author has received):

Line 84-92 – the author writes ‘These previous studies investigating hypertensive patients’ knowledge levels in other Saudi regions cannot be generalized to other Saudi regions, such as the Jazan region, where a higher proportion of people suffer from hypertension and its complications and where the health system is overwhelmed by emergencies and cases of communicable and genetic diseases’ – can you conform whether study with numbered reference 15 (by Alshammari et al) was conducted in other region of Saudi Arabia? Or the sample was taken from the whole of Saudi Arabia? If the latter is true, then you will need to refine the statement of your evidence gap.

Response: We added further information related to study settings (by Alshammari et al) to be “conducted in various regions of Saudi Arabia”. Page 5, line 88. We also refined the statement of our evidence gap to be “As a result of the limited data, there is a need to understand the knowledge in general and specific areas concerning hypertension in this particular region of Saudi Arabia.”. Page 5, lines 97-98.

Reviewer #1

Minor comments:

1.1 The manuscript is technically sound, the data support the conclusions and presented in an intelligible fashion.

Response: Thank you for your appreciation.

1.2 In the title its very important to indicate the type of the study. No need to mention the region of the study to enhance the global significant. You can add the Country name. 

Response: Thank you for your remarkable comment. We have incorporated the reviewer’s suggestion into the title to be “Assessing knowledge about hypertension and identifying predictors of inadequate knowledge in Saudi Arabia: A cross-sectional study”. Page 1, line 2.

1.3 The conclusion section of the abstract should be rephrased according to the main findings of the study.

Response: Thank you for your valuable comment. We have omitted “Therefore, to improve patients' knowledge and enhance public health practices in Jazan, involving public health educators in primary health care services could improve patients' knowledge about hypertension and its complications and reduce the burden on family physicians in the Jazan region shown in the literature. Further studies are required to determine the knowledge level of hypertensive patients with severe complications such as cerebral stroke.” and rephrased the conclusion section in the abstract in line with the main findings to be “Diet, medical treatment, disease definition, drug compliance, and complications were subsequently the least knowledgeable subdimensions among the study population. Therefore, these subdimensions should be prioritized when planning hypertension educational interventions and during follow-up sessions, especially for patients of younger age groups and those with lower educational levels.”. Page 3, lines 45-49.

1.4 The sampling technique and the online questionnaire preclude the presentation of whole area and generalizability of the results. This must be added to the limitation section.

Response: We have omitted the statement “Second, using a nonprobability sampling strategy affected the study’s external validity, which could limit its generalizability” and incorporated the reviewer’s suggestion to be “Second, the sampling technique and the online questionnaire preclude the presentation of the whole area and the generalizability of the results.”. Page 19, lines 316-318.

Reviewer #2

Minor comments (recommendations):

2.1 It is well written and technically sound. No issues of dual publications and research ethics. Although it attempted to address this important topic there are some critical issues that warrant further attention. These would improve the research's quality and robustness, making it more academic journal-worthy. 

Response: Thank you for your appreciation.

2.2 Title and Abstract: Text clarity, conciseness and structure can be improved. 

Response: According to reviewer recommendations, we have improved the text clarity, conciseness, and structure in the title and abstract. Page 1, line 2, page 2, lines 25-27, line 30, lines 34-36, and page 3, lines 45-49.

2.3 Literature assessment: A more thorough assessment would strengthen the study's scientific foundation. This would help understand the present state of knowledge and the research gaps the project attempts to address. 

Response: Thank you for your suggestion. We have added further information related to the literature assessment to the introduction section to strengthen the scientific foundation. Page 5, lines 89-91, and lines 97-98.

2.4 Methods: More detailed information about the tool for data would be helpful. 

Response: We have added detailed information in the material and methods section; the data collection instrument to be “which was developed by Erkoc et al. in English to assess Turkish adults’ knowledge about hypertension and its management using 22 items that are distributed across six specific areas called subdimensions, namely: disease definition, medical treatment, drug compliance, lifestyle, diet, and complications”, page 8, lines 153-156, and we also added further explanation as the following “an addition of one item to the disease definition subdimension, and it was added to assess participants’ knowledge regarding the increased isolated systolic pressure as a form of hypertension. Also, three items were added to the lifestyle subdimension to evaluate patients’ understanding of the role of salt intake, obesity/overweight, and physical activity as essential factors for managing hypertension.”. Page 8, lines 158-162.

2.5 Discussion: Attention and conciseness can be improved by emphasizing the most important facts and recommendations.

Response: Thank you for your recommendation. The most important facts and recommendations were emphasized in the discussion section. For example, we discussed our findings related to lower level of knowledge that were observed at the overall scale and each subdimension. In addition, we have discussed the uncontrolled hypertension and identified regional structural indicators necessary for dealing with hypertension complications, the co-existence of other prevalent health problems, hypertension misleading information, lack of health professionals, and low adherence to the hypertension guidelines. We also draw the attention of readers and policy-makers to involve health educators in such situations to improve hypertension management.

---

## [Editor Report · Decision Letter 1]

16 Feb 2024

Assessing knowledge about hypertension and identifying predictors of inadequate knowledge in Saudi Arabia: A cross-sectional study

PONE-D-23-25700R1

Dear Ajiad Alhazmi,

We’re pleased to inform you that your manuscript has been judged scientifically suitable for publication and will be formally accepted for publication once it meets all outstanding technical requirements.

Kind regards,

Isaac Amankwaa, Ph.D.

Guest Editor

PLOS ONE
---

## [Editor Report · Acceptance letter]

7 Mar 2024

PONE-D-23-25700R1 

PLOS ONE

Dear Dr. Alhazmi, 

I'm pleased to inform you that your manuscript has been deemed suitable for publication in PLOS ONE. Congratulations! Your manuscript is now being handed over to our production team.

Kind regards, 

on behalf of

Dr. Isaac Amankwaa 

Guest Editor

PLOS ONE